# Learning Performance-oriented Control Barrier Functions Under Complex Safety Constraints and Limited Actuation

**Lakshmideepakreddy Manda**
Department of Electrical and Computer Engineering
Johns Hopkins University
United States
lmanda1@jhu.edu

**Shaoru Chen**
Microsoft Research
New York, United States
shaoruchen@microsoft.com

**Mahyar Fazlyab**
Department of Electrical and Computer Engineering
Johns Hopkins University, United States
mahyarfazlyab@jhu.edu

**Abstract:** Control Barrier Functions (CBFs) provide an elegant framework for constraining nonlinear control system dynamics to remain within an invariant subset of a designated safe set. However, identifying a CBF that balances performance—by maximizing the control invariant set—and accommodates complex safety constraints, especially in systems with high relative degree and actuation limits, poses a significant challenge. In this work, we introduce a novel self-supervised learning framework to comprehensively address these challenges. Our method begins with a Boolean composition of multiple state constraints that define the safe set. We first construct a smooth function whose zero superlevel set forms an inner approximation of this safe set. This function is then combined with a smooth neural network to parameterize the CBF candidate. To train the CBF and maximize the volume of the resulting control invariant set, we design a physics-informed loss function based on a Hamilton-Jacobi Partial Differential Equation (PDE). We validate the efficacy of our approach on a 2D double integrator (DI) system and a 7D fixed-wing aircraft system (F16).

## 1   Introduction

CBFs are a powerful tool to enforce safety constraints for nonlinear control systems [1], with many successful applications in autonomous driving [2], UAV navigation [3], robot locomotion [4], and safe reinforcement learning [5]. For control-affine nonlinear systems, CBFs can be used to construct a convex quadratic programming (QP)-based safety filter deployed online to safeguard against potentially unsafe control commands. The induced safety filter, denoted as CBF-QP, corrects the reference controller to remain in a safe *control invariant set*.

While Control Barrier Functions (CBFs) provide an efficient method to ensure safety, finding such functions can be challenging. Specifically, there is no guarantee that the resulting safety filter will remain feasible throughout operation. This is primarily because the "model-free" construction of the filter only incorporates the constraints. Complex constraints, high relative degree, and bounded actuation exacerbate the challenge of ensuring feasibility. Various techniques have been proposed to address these challenges such

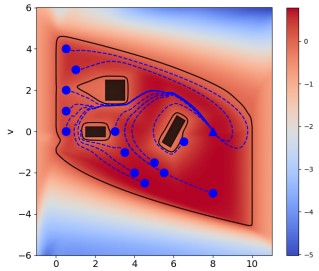

Figure 1: Illustration of the learned CBF-QP filtering many initializations of PID reference control on the DI system. The CBF zero contour drawn on its value heatmap bounds the learned control invariant set.

8th Conference on Robot Learning (CoRL 2024), Munich, Germany.

as CBF composition for complex constraints [6, 7, 8, 9], higher-order CBFs for high relative degree [10] [11], and integral CBFs [12] for limited actuation. Despite significant progress, these approaches can make the filter overly restrictive, thus limiting performance.

**Contributions** We propose a novel self-supervised learning framework for CBF synthesis that systematically addresses all the above challenges. First, we handle complex safety constraints and high relative degree in CBF synthesis by encoding the safety constraints into the CBF parameterization with minimal conservatism. Second, we design a physics-informed training loss function based on Hamilton-Jacobi (HJ) reachability analysis [13] to satisfy bounded actuation while maximizing the learned control invariant set volume. We evaluate our method on the double-integrator and the high-dimensional fixed-wing aircraft system and demonstrate that the proposed method effectively learns a performant CBF even with complex safety constraints. We call the proposed framework Physics-informed Neural Network (PINN)-CBF [14].

## 1.1 Related work

**Complex safety constraints** For a safe set described by Boolean logical operations on multiple constraints, [6] composes multiple CBFs accordingly through the non-smooth min/max operators.[7] introduces smooth composition of logical operations on constraints, which was later extended to simultaneously handle actuation and state constraints [8][9] using integral CBFs [12]. [15] proposes an algorithmic way to create a single smooth CBF arbitrarily composing both min and max operators. Such smooth bounds have been used in changing environments [16]. Notably, [17] ensures the input constrained feasibility of the CBF condition while composing multiple CBFs.

**High-order CBF** High-order CBF (HOCBF) [10] and exponential CBF [11] are systematicly approach CBF construction when the safety constraints have high relative degree. However, controlling the conservatism of these approaches is a challenge. To reduce conservatism or improve the performance of HOCBF, various learning frameworks have been proposed [18, 19, 20] that allow tuning of the class $\mathcal{K}$ functions used in the CBF condition.

**Learning CBF with input constraints** Motivated by the difficulty of hand-designing CBFs, learning-based approaches building on the past [21] have emerged in recent years as an alternative [22, 23, 24, 25]. Liu et al. [26] explicitly consider input constraints in learning a CBF by finding counterexamples on the $0$-level set of the CBF for training, while Dai et al. [27] propose a data-efficient prioritized sampling method. [28] explores adaptive sampling for training HJB NN models favoring sharp gradient regions. Drawing tools from reachability analysis, the recent work [29] iteratively expands the volume of the control invariant set by learning the policy value function and improving the performance through policy iteration. Similarly, Dai et al. [30] expand conservative hand-crafted CBFs by learning on unsafe and safe trajectories, and Qin et al. [31] applies actor-critic RL framework to learn a CBF value function while [32] learns a control policy and CBF together for black-box constraints and dynamics. Our method differs by being controller-independent and basing its learning objective on the HJ partial differential equation (PDE), which defines the maximal control invariant set, without requiring trajectory training data.

**HJ reachability-based methods** The value functions in HJ reachability analysis have been extended to construct control barrier-value functions (CBVF) [33] and control Lyapunov-value functions [34], which can be computed using existing toolboxes [35]. Tokens et al. [36] further apply such tools to refine existing CBF candidates, and Bansal et al. [37] learn neural networks solution to a HJ PDE for reachability analysis. Of particular interest to our work is the CBVF, which is close to the CBF formulation and provides a characterization of the viability kernel. Our work aims to learn a neural network CBF from data without using computational tools based on spatial discretization.

**Notation** An extended class $\mathcal{K}$ function is a function $\alpha : (-b, a) \mapsto \mathbb{R}$ for some $a, b > 0$ that is strictly increasing and satisfies $\alpha(0) = 0$. We denote $L_r^+(h) = \{x \in \mathbb{R}^n \mid h(x) \geq r\}$ as the $r$-superlevel set of a continuous function $h(x)$. The positive and negative parts of a number $a \in \mathbb{R}$ are denoted by $(a)_+ = \max(a, 0)$ and $(a)_- = \min(a, 0)$, respectively. Let $\wedge, \vee, \neg$ denote the logical operations of conjunction, disjunction, and negation, respectively. For two statements $A$ and $B$, we have $\neg(A \wedge B) = (\neg A) \vee (\neg B)$ and $\neg(A \vee B) = (\neg A) \wedge (\neg B)$.

## 2 Background and Problem Statement

Consider a continuous time control-affine system:

$$\dot{x} = f(x) + g(x)u, \quad u \in \mathcal{U}, \tag{1}$$

where $\mathcal{D} \subseteq \mathbb{R}^n$ is the domain of the system, $x \in \mathcal{D}$ is the state, $u \in \mathcal{U} \subset \mathbb{R}^m$ is the control input, $\mathcal{U} \subseteq \mathbb{R}^m$ denotes the control input constraint. We assume that $f \colon \mathcal{D} \to \mathbb{R}^n$, $g \colon \mathcal{D} \to \mathbb{R}^{n \times m}$ are locally Lipschitz continuous and $\mathcal{U}$ is a convex polyhedron. We denote the solution of (1) at time $t \geq 0$ by $x(t)$. Given a set $\mathcal{X} \subseteq \mathcal{D}$ that represents a safe subset of the state space, the general objective of safe control design is to find a control law $\pi(x)$ that renders $\mathcal{X}$ invariant under the closed-loop dynamics $\dot{x} = f(x) + g(x)\pi(x)$, i.e., if $x(t_0) \in \mathcal{X}$ for some $t_0 \geq 0$, then $x(t) \in \mathcal{X}$ for all $t \geq t_0$. A general approach to solving this problem is through control barrier functions.

### 2.1 Control Barrier Functions

Suppose the safe set is defined by the 0-superlevel set of a smooth function $c(\cdot)$ such that $\mathcal{X} = \{x \mid c(x) \geq 0\}$. For $\mathcal{X}$ to be control invariant, the boundary function $c(\cdot)$ must satisfy $\max_{u \in \mathcal{U}} \dot{c}(x, u) \geq 0$ when $c(x) = 0$ by Nugomo's theorem [38]. However, since $c(\cdot)$ does not necessarily satisfy the condition, we settle with finding a control invariant set contained in $\mathcal{X}$ through CBFs.

**Definition 1** (Control barrier function). *Let $\mathcal{S} := L_0^+(h) \subseteq \mathcal{X} \subseteq \mathcal{D}$ be the 0-superlevel set of a continuously differentiable function $h \colon \mathcal{D} \to \mathbb{R}$. Then $h(\cdot)$ is a control barrier function for system (1) if there exists an extended class $\mathcal{K}$ function $\alpha$ such that*

$$\sup_{u \in \mathcal{U}} \{L_f h(x) + L_g h(x)u + \alpha(h(x))\} \geq 0, \forall x \in \mathcal{D}, \tag{2}$$

*where $L_f h(x) = \nabla h(x)^\top f(x)$ and $L_g h(x) = \nabla h(x)^\top g(x)$ are the Lie derivatives of $h(x)$.*

Given a CBF $h(\cdot)$, the non-empty set of point-wise safe control actions is given by any locally Lipschitz continuous controller $\pi(x) \in \mathcal{K}_{\mathrm{cbf}}(x) = \{u \in \mathcal{U} \mid L_f h(x) + L_g h(x)u + \alpha(h(x)) \geq 0\}$ renders the set $\mathcal{S}$ forward invariant for the closed-loop system, which enables the construction of a minimally-invasive safety filter:

$$\pi(x) := \operatorname*{argmin}_{u} \|u - u_r(x)\|_2^2 \quad \text{subject to} \quad L_f h(x) + L_g h(x)u + \alpha(h(x)) \geq 0, \ u \in \mathcal{U}, \tag{3}$$

where $u_r(x)$ is any given reference but potentially unsafe controller. Under the assumption that $\mathcal{U}$ is a polyhedron, $\mathcal{K}_{\mathrm{cbf}}(x)$ is also a polyhedral set and problem (3) becomes a convex QP.

### 2.2 Problem Statement

While the CBF-QP filter is minimally invasive and guarantees $x(t) \in L_0^+(h) \subseteq \mathcal{X}$ for all time, a small $L_0^+(h)$ essentially limits the ability of the reference control to execute a task. To take the performance of the reference controller into account, we consider the following problem.

**Problem 1** (Performance-Oriented CBF). *Given the input-constrained system (1) and a safe set $\mathcal{X}$ defined by complex safety constraints (to be specified in Section 3.1), synthesize a CBF $h(\cdot)$ with an induced control invariant set $\mathcal{S} = L_0^+(h)$ such that (i) $\mathcal{S} \subseteq \mathcal{X}$ and (ii) the volume of $\mathcal{S}$ is maximized. Formally, this problem can be cast as an infinite-dimensional optimization problem:*

$$\underset{h \in \mathcal{H}}{\text{maximize}} \quad \text{volume}(L_0^+(h)) \quad (\texttt{performance}) \tag{4}$$

$$\text{subject to} \quad L_0^+(h) \subseteq \mathcal{X} \quad (\texttt{safety})$$

$$\sup_{u \in \mathcal{U}} \{L_f h(x) + L_g h(x)u + \alpha(h(x))\} \geq 0, \ \forall x \in \mathcal{D} \ (\texttt{control invariance})$$

*where $\mathcal{H}$ is the class of scalar-valued continuously differentiable functions.*

## 3 Proposed Method

In this section, we present our three-step method of learning PINN-CBF :

**(S1) Composition of complex state constraints:** Given multiple state constraints composed by Boolean logic describing the safe set $\mathcal{X}$, we equivalently represent $\mathcal{X}$ as a zero super level set of a single non-smooth function $c(\cdot)$, i.e., $\mathcal{X} = L_0^+(c) = \{x \mid c(x) \geq 0\}$.

**(S2) Inner approximation of safe set:** Given the constraint function $c(\cdot)$ obtained from the previous step, we derive a smooth minorizer $\underline{c}(\cdot)$ of $c(\cdot)$, i.e., $c(x) \geq \underline{c}(x)$ for all $x \in \mathcal{D}$, s.t. $L_0^+(\underline{c}) \subseteq \mathcal{X}$.

**(S3) Learning performance-oriented CBF:** To approximate the largest control invariant subset of $L_0^+(\underline{c})$, we design a training loss function based on control barrier-value functions and HJ PDE [33]. We propose a parameterization of the CBF and a sampling strategy exploiting the structure of the PDE.

## 3.1 Composition of Complex State Constraints

Suppose we are given $N$ sets $\mathcal{S}_i := L_0^+(s_i) = \{x \mid s_i(x) \geq 0\}$ where each $s_i : \mathbb{R}^n \mapsto \mathbb{R}$ is continuously differentiable and the safe set $\mathcal{X}$ is described by logical operations on $\{\mathcal{S}_i\}_{i=1}^N$. Since all Boolean logical operations can be expressed as the composition of the three fundamental operations conjunction, disjunction, and negation [6, 15], it suffices to only demonstrate the set operations shown below:

1. Conjunction: $x \in \mathcal{S}_i$ AND $x \in \mathcal{S}_j \Leftrightarrow x \in \mathcal{S}_i \cap \mathcal{S}_j = \{x \mid \tilde{s}(x) := \min(s_i(x), s_j(x)) \geq 0\}$.
2. Disjunction: $x \in \mathcal{S}_i$ OR $x \in \mathcal{S}_j \Leftrightarrow x \in \mathcal{S}_i \cup \mathcal{S}_j = \{x \mid \tilde{s}(x) := \max(s_i(x), s_j(x)) \geq 0\}$.
3. Negation: NOT $x \in \mathcal{S}_i \Leftrightarrow x \in \mathcal{S}_i^{\complement} = \{x \mid \tilde{s}(x) := -s_i(x) \geq 0\}$ (complement of $\mathcal{S}_i$).

The conjunction and disjunction of two constraints can be exactly expressed as one constraint composed through the $\min$ and $\max$ operator, respectively. Furthermore, negating a constraint $s_i(x) \geq 0$ only requires flipping the sign of $s_i$. These logical operations enable us to capture complex geometries and logical constraints as illustrated in the following two examples.

**Example 1** (Complex geometric sets). *Consider $x = [x_1 \ x_2]^\top \in \mathbb{R}^2$ and two rectangular obstacles given by $\mathcal{O}_i := \{x \mid \begin{bmatrix} a_i \\ b_i \end{bmatrix} \leq x \leq \begin{bmatrix} c_i \\ d_i \end{bmatrix}\}$ with $i = 1, 2$. The union of the two rectangular obstacles $\mathcal{O}_1 \cup \mathcal{O}_2$ is a nonconvex set. Define the following functions*

$$s_1(x) = -x_1 + c_1, s_2(x) = x_1 - a_1, s_3(x) = -x_2 + d_1, s_4(x) = x_2 - b_1,$$
$$s_5(x) = -x_1 + c_2, s_6(x) = x_1 - a_2, s_7(x) = -x_2 + d_2, s_8(x) = x_2 - b_2,$$

*and let $c(x) = \max(\min(s_1(x), s_2(x), s_3(x), s_4(x)), \min(s_5(x), s_6(x), s_7(x), s_8(x)))$. Then, we have $L_0^+(c) = \mathcal{O}_1 \cup \mathcal{O}_2$.*

**Example 2** (Logical constraints). *Consider the three constraints $s_i(x) \geq 0, i = 1, 2, 3$, at least two of which must be satisfied. This specification is equivalent to the constraint $c(x) \geq 0$, where*

$$c(x) = \max(\min(s_1(x), s_2(x)), \min(s_2(x), s_3(x)), \min(s_1(x), s_3(x))).$$

In summary, by composing the $\min$ and $\max$ operators, we can construct a level-set function $c : \mathbb{R}^n \mapsto \mathbb{R}$ such that $x \in \mathcal{X} \Leftrightarrow c(x) \geq 0$, i.e., $\mathcal{X} = L_0^+(c)$. Being an exact description of $\mathcal{X}$, however, $c(\cdot)$ is not smooth. Next, we find a smooth lower bound of $c(\cdot)$ that facilitates CBF design.

## 3.2 Inner Approximation of Safe Set

To find a smooth lower bound for $\underline{c}(\cdot)$, we utilize its compositional structure. We bound the $\max$ operators using the log-sum-exponential function as follows [15] ($\min$ follows similarly):

$$\frac{1}{\beta} \log(\sum_{i=1}^M e^{\beta s_i}) - \frac{\log(M)}{\beta} \leq \max(s_1, \cdots, s_M) \leq \frac{1}{\beta} \log(\sum_{i=1}^M e^{\beta s_i}), \tag{5}$$

with $\beta > 0$. As $\beta \to \infty$, these bounds can be made arbitrarily accurate. We note that both the lower and upper bounds in (5) are smooth and strictly increasing in each input $s_i$. Therefore, to obtain a lower bound $\underline{c}(x)$ of $c(x)$, it suffices only to compose the lower bounds on each $\min$ and $\max$ function. In Figure 2 we show the effect of $\beta$ on the resulting inner approximation.

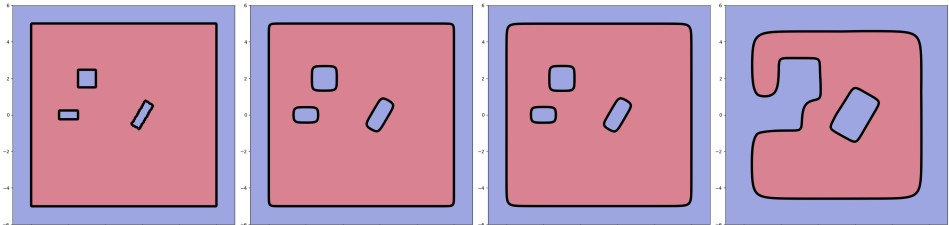

Figure 2: Effect of the smoothing parameter $\beta$ on the resulting inner approximation. $\beta \in \{+\infty, 10, 5, 2\}$.

### 3.3 Learning PINN-CBF

The smooth function $\underline{c}(\cdot)$, whose 0-superlevel set provides an inner approximation of the safe set $\mathcal{X}$, is not necessarily a CBF. Thus, we aim to find the "closest" CBF approximation of $\underline{c}(\cdot)$. We learn our Neural Network (NN) model using a Hamilton-Jacobi (HJ) PDE from reachability analysis whose infinite-time horizon solution precisely characterizes the CBF maximizing the volume of $L_0^+(h)$. Our neural network model is trained by minimizing the PDE residual, grounding the method in physics, and justifying its name (PINN)-CBF.

**Hamilton-Jacobi PDE for reachability**   Consider the dynamics (1) in the time interval $[t, 0]$, where $t \leq 0$ and $x$ are the initial time and state, respectively. Define $\mathcal{U}_{[t,0]}$ as the set of Lebesgue measurable functions $u \colon [t, 0] \to \mathcal{U}$. Let $\psi(s) := \psi(s; x, t, u(\cdot)) : [t, 0] \mapsto \mathbb{R}^n$ denote the unique solution of (1) given $x$ and $u(\cdot) \in \mathcal{U}_{[t,0]}$. Given a bounded Lipschitz continuous function $\ell : \mathcal{D} \mapsto \mathbb{R}$, the viability kernel of $L_0^+(\ell) = \{x \mid \ell(x) \geq 0\}$ is defined as

$$\mathcal{V}(t) := \{x \in L_0^+(\ell) \mid \exists u(\cdot) \in \mathcal{U}_{[t,0]} \text{ s.t. } \forall s \in [t, 0], \psi(s) \in L_0^+(\ell)\}, \tag{6}$$

which is the set of all initial states within $L_0^+(\ell)$ from which there exists an admissible control signal $u(\cdot)$ that keeps the system trajectory within $L_0^+(\ell)$ during the time interval $[t, 0]$. Solving for the viability kernel can be posed as an optimal control problem, where $\mathcal{V}(t)$ can be expressed as the superlevel set of a value function called control barrier-value function (CBVF).

**Definition 2** (CBVF [33])**.** *Given a discount factor $\gamma \geq 0$, the control barrier-value function $B_\gamma :$ $\mathcal{D} \times (-\infty, 0] \mapsto \mathbb{R}$ is defined as $B_\gamma(x, t) := \max\limits_{u(\cdot) \in \mathcal{U}_{[t,0]}} \min\limits_{s \in [t,0]} e^{\gamma(s-t)} \ell(\psi(s)).$*

For $t \leq 0$, we have $\mathcal{V}(t) = \{x \mid B_\gamma(x, t) \geq 0\}$ [33, Proposition 2]. Additionally, $B_\gamma$ is the unique Lipschitz continuous viscosity solution of the following HJ PDE with terminal condition $B_\gamma(x, 0) = \ell(x)$ [33, Theorem 3]:

$$\min\left\{\ell(x) - B_\gamma(x, t), \frac{\partial}{\partial t} B_\gamma(x, t) + \max_{u \in \mathcal{U}} \nabla_x B_\gamma(x, t)^\top (f(x) + g(x)u) + \gamma B_\gamma(x, t)\right\} = 0. \tag{7}$$

Under mild assumptions of Lipschitz continuous dynamics and $\ell(x)$ being a signed distance function, $B_\gamma(x, t)$ is differentiable almost everywhere [39]. Furthermore, taking $t \to -\infty$, the steady state solution $B_\gamma(x) := B_\gamma(x, -\infty)$ gives us the *maximal* control invariant set contained in $L_0^+(\ell)$ [33, Section II.B]:

$$\mathcal{N}(B_\gamma, \ell) := \min\left\{\ell(x) - B_\gamma(x), \max_{u \in \mathcal{U}} L_f B_\gamma(x) + L_g B_\gamma(x)u + \gamma B_\gamma(x)\right\} = 0, \ \forall x \in \mathcal{D} \tag{8}$$

We will leverage the above PDE to learn a CBF $h$ whose zero superlevel set approximates the maximal control invariant set.

**CBF parameterization**   In the context of our problem, we are interested in the viability kernel of $L_0^+(\underline{c}(x))$, where $\underline{c}$ is the smoothed composition of the constraints. Thus, our goal is to learn a CBF $h$ that satisfies the PDE $\mathcal{N}(h, \underline{c}) = 0$. To this end, we parameterize the CBF candidate as

$$h_\theta(x) = \underline{c}(x) - \delta_\theta(x), \tag{9}$$

where $\delta_\theta \colon \mathcal{D} \to \mathbb{R}_{\geq 0}$ is a non-negative continuously differentiable function approximator with parameters $\theta$. In this paper, we parameterize $\delta_\theta$ in the form of a multi-layer perceptron:

$$\delta_\theta(x) = \sigma_+(W_L z_L + b_L), \ z_{k+1} = \sigma(W_k z_k + b_k), k = 0, \cdots, L-1, \ z_0 = x, \tag{10}$$

where $\{W_k, b_k\}_{k=0}^K$ are the learnable parameters, $\sigma(\cdot)$ is a smooth activation such as the ELU, Tanh, or Swish functions [40], and $\sigma_+(\cdot)$ is a smooth function with non-negative outputs, i.e., $\sigma_+(r) \geq 0, \forall r \in \mathbb{R}$. We choose $\sigma_+(\cdot)$ as the Softplus function, $\sigma_+(r) = \frac{1}{\beta} \log(1 + \exp(\beta r)), \ \beta > 0$.

The parameterization (9) offers several advantages over using a standard MLP model for $h_\theta$. First, it accelerates learning by enforcing the non-negativity constraint $\underline{c}(x) \geq h(x)$ in (8) by design. Second, complex constraints are directly integrated into the system via $\underline{c}(x)$, simplifying the learning process. Importantly, $h_\theta(x)$ is automatically non-positive in regions containing obstacles, allowing the learning process to focus elsewhere.

In summary, the parameterization in (9) ensures that $h_\theta(\cdot)$ is smooth and satisfies $h_\theta(x) \leq \underline{c}(x) \leq c(x)$ for all $x \in \mathcal{D}$, implying that $L_0^+(h_\theta) \subseteq L_0^+(\underline{c}) \subseteq L_0^+(c) = \mathcal{X}$ by construction.

**Initialization** We propose a specialized initialization scheme by letting $W_L, b_L = 0$, initializing the candidate $h_\theta$ with $\underline{c}$. This initialization favors finding the closest CBF to $\underline{c}(x)$ during training. Setting only the last layer to zero allows training to break symmetry and retain gradient flow.

**Training loss and sampling distribution** Given the parameterization (9), we now propose to train $h_\theta$ to approximately satisfy the steady-state HJ PDE. This leads to the following physics-informed risk minimization

$$\underset{\theta}{\text{minimize}} \quad J(\theta) = \mathbb{E}_{x \sim \mu} \left[ \mathcal{L}_{\text{HJ}}(x; \theta, \gamma) + \lambda \mathcal{L}_{\text{CBF}}(x; \theta, \gamma)^2 \right], \quad \lambda \geq 0. \tag{11a}$$

$$\mathcal{L}_{\text{HJ}}(x; \theta, \gamma) = \left( \min \left\{ \underline{c}(x) - h_\theta(x), \ \max_{u \in \mathcal{U}} L_f h_\theta(x) + L_g h_\theta(x) u + \gamma h_\theta(x) \right\} \right)^2 \tag{11b}$$

$$\mathcal{L}_{\text{CBF}}(x; \theta, \gamma) = \max \left\{ - \max_{u \in \mathcal{U}} L_f h_\theta(x) - L_g h_\theta(x) u - \gamma h_\theta(x), 0 \right\}. \tag{11c}$$

The first loss $\mathcal{L}_{\text{HJ}}(x; \theta, \gamma)$ is the square of the PDE residual to increase the volume of $L_0^+(h_\theta)$, while the second loss $\mathcal{L}_{\text{CBF}}(x; \theta, \gamma)$ enforces the CBF condition. We note that the latter is implicitly present in the PDE but not enforced by the corresponding loss $\mathcal{L}_{\text{HJ}}(x; \theta, \gamma)$.

Finally, $\mu$ represents a sampling distribution over $L_0^+(\underline{c})$, meaning *no learning is required* within obstacle regions. Efficient sampling in these regions can be achieved using intelligent sampling methods like envelope rejection [41] or the random walk Metropolis-Hastings (RWMH) [42], both of which are well-suited for cluttered environments. Notably, the training process remains self-supervised, as only domain points are collected for learning.

In the following, we demonstrate that jointly enforcing $\mathcal{L}_{\text{HJ}}$ and $\mathcal{L}_{\text{CBF}}$ together with intelligent sampling (IS) improves the performance and safety of the learned PINN-CBF. We note that for polyhedral actuation constraint sets, the supremum over $u$ is achieved at one of the vertices, yielding a closed-form expression. We summarize our self-supervised training method in Algorithm 1.

---

**Algorithm 1** Training PINN-CBFs

---

**Input:** Constraints $\{c_i\}$, smoothing parameter $\beta > 0$, discount factor $\gamma > 0$, regularization parameter $\lambda \geq 0$
**Output:** Modules $\delta_\theta(\cdot)$ and $\underline{c}(\cdot)$ s.t. $h_\theta = \underline{c}(\cdot) - \delta_\theta(\cdot)$

**1.** Following sections 3.1 and 3.2, compose constraints $\{c_i\}$ using $\beta$ to form $\underline{c}$.
**2.** Sample training data, $X = \{x_i \in L_0^+(\underline{c})\}_{i=1}^N$.
**3.** Initialize weights and biases of the model $\delta_\theta$ randomly with $W_L, b_L = 0$.
**4.** ERM: $\underset{\theta}{\text{minimize}} \quad \hat{J}(\theta) = \frac{1}{N} \sum_{i=1}^N \left[ \mathcal{L}_{\text{HJ}}(x_i; \theta, \gamma) + \lambda \mathcal{L}_{\text{CBF}}(x_i; \theta, \gamma)^2 \right]$

---

# 4 Experiments

We demonstrate the efficacy of PINN-CBF on a 2D double integrator system and a 7D fixed-wing aircraft system. Experiments are performed on Google Colab, where the PINN-CBF training is performed on a T4 GPU with 15GB RAM and its validation on a CPU with 52GB RAM. The code is available at https://github.com/o4lc/PINN-CBF.

## 4.1 Experiment Setup

**Double Integrator**  First, we consider the double integrator benchmark $\dot{x} = v \quad \dot{v} = u$, with $x \in \mathbb{R}$ denoting the position and $v \in \mathbb{R}$ denoting the velocity. The action $u \in \mathbb{R}$ represents the acceleration of the system and directly operates on $v$.

We generate a complex obstacle configuration consisting of rotated rectangles and unit walls bordering the state space $\mathcal{D} = \{(x, v) \mid -1 \leq x \leq 11, -6 \leq v \leq 6\}$ shown in Fig. 2. Then, we parameterize PINN-CBF following Section 3.3 and train it with $N = 10^4$ uniform samples. At each state in the training data, we compute the values and gradient of $\underline{c}$ using JAX [43] flexible automatic-differentiation and hardware acceleration. The acceleration $u$ is bounded by $u \in [-1, 1]$. The nominal controller is given by a PID controller meant to stabilize the system to a target position (see Appendix A.1).

**Fixed-Wing Aircraft**  The Dubins fixed-wing Plane system has 7D states and 3D limited actuation control. Its full dynamics are shown in Appendix A.2. Sampling and training are done as before except with $N = 10^6$ to compensate for a higher dimensional space $\mathcal{D} = \{(n, e, d, \phi, \theta, \psi, V_T) \mid -6 \leq n, e \leq 6, -1 \leq d \leq 11, -2\pi \leq \phi, \theta, \psi \leq 2\pi, 0.5 \leq V_T \leq 2\}$. Although $10^6 \ll (\sqrt{10^4})^7 = 10^{14}$, as suggested by naive dimensional scaling, we aim to demonstrate how learning can overcome the curse of dimensionality. The action $u = \{A_T, p, q\}$ is bounded in magnitude by $|u| \leq [10.5, 1, 1]$. Rectangular obstacles occur in $n, e, d$. We chose a nominal trajectory resembling takeoff to evaluate the PINN-CBF filter against relevant baselines on collision avoidance.

## 4.2 Results

**Double-Integrator**  We evaluate the combination of loss functions (11b) and (11c) with intelligent sampling (IS) from Algorithm 1, as shown in Table 1. All proposed ablations and baseline methods use the same architecture (9) for a fair comparison, accommodating complex input-space constraints. The MLP for $\delta_\theta$ has layer widths of $2 - 50 - 50 - 1$, while DeepReach [37] includes time as an additional input and is applied over the full horizon with $t = 0$.

Using three metrics, We validate at $10^5$ grid samples covering the effective obstacle-free space $L_0^+(\underline{c})$. The residual error $|\mathcal{N}(h, \underline{c})|$ denotes how close the learned CBF is to the maximal-invariant solution of the HJ PDE, $\mathcal{L}_{\text{CBF}}$ measures safety violation, and $\text{Vol}(L_0^+(h_\theta))/\text{Vol}(L_0^+(\underline{c}))$ represents the volume ratio between the learned invariant set and the smoothed safe set, which serves as an upper bound enforced by the architecture.

| Method | Mean $\|\mathcal{N}(h, \underline{c})\|$ over $L_0^+(\underline{c})$ | Mean $\mathcal{L}_{\text{CBF}}$ over $L_0^+(\underline{c})$ | $\text{Vol}(L_0^+(h_\theta))/\text{Vol}(L_0^+(\underline{c}))$ |
|---|---|---|---|
| $\mathcal{L}_{\text{HJ}}$ | 0.026 | 0.016 | 0.566 |
| $\mathcal{L}_{\text{HJ}} + \lambda \mathcal{L}_{\text{CBF}}^2$ | 0.026 | 0.015 | 0.547 |
| $\mathcal{L}_{\text{HJ}} + \lambda \mathcal{L}_{\text{CBF}}^2$ (IS) | 0.022 | 0.011 | 0.619 |
| $\mathcal{L}_{\text{CBF}}^2$ [22] | 2.002 | $8.057 \times 10^{-5}$ | 0.0 |
| NCBF [26] | 0.178 | 0.177 | 1.0 |
| DeepReach [37] | 0.206 | 0.195 | 0.984 |

Table 1: Comparison of $\mathcal{L}_{\text{HJ}}$ methods with our ablations above baselines for the 2D Double-Integrator system.

In the ablations, scalarization with $\lambda = 0.2$ reduces safety violations but decreases volume. In contrast, intelligent sampling lowers residuals and safety violations while increasing volume, suggesting it mitigates the tradeoff of scalarization. Among the baselines, directly enforcing the CBF condition (2) with $\mathcal{L}_{\text{CBF}}^2$ results in a highly safe but ineffective filter with zero volume. NCBF and DeepReach achieve high volume but with significantly higher safety errors. Next, we will examine how these methods perform on a more complex fixed-wing plane system.

**Fixed-Wing Plane** The same comparisons conducted for the Double-Integrator are also performed with the following adjustments: all methods utilize a $7-100-100-1$ MLP architecture. DeepReach includes an additional input for time. Additionally, validation is carried out on $10^7$ uniformly random sampled states instead of grid samples.

| Model | Mean $\|\mathcal{N}(h, \underline{c})\|$ over $L_0^+(\underline{c})$ | Mean $\mathcal{L}_{\text{CBF}}$ over $L_0^+(\underline{c})$ | Volume of $L_0^+(h_\theta)/L_0^+(\underline{c})$ |
|---|---|---|---|
| $\mathcal{L}_{\text{HJ}}$ | 0.345 | 0.088 | 0.707 |
| $\mathcal{L}_{\text{HJ}} + \lambda\mathcal{L}_{\text{CBF}}^2$ | 0.345 | 0.082 | 0.702 |
| $\mathcal{L}_{\text{HJ}} + \lambda\mathcal{L}_{\text{CBF}}^2$ (IS) | 0.339 | 0.081 | 0.700 |
| $\mathcal{L}_{\text{CBF}}^2$ [22] | 5.794 | $2.731 \times 10^{-4}$ | $8.400 \times 10^{-6}$ |
| NCBF [26] | 0.741 | 0.240 | 0.579 |
| DeepReach [37] | 0.543 | 0.500 | 0.951 |

Table 2: Comparison of $\mathcal{L}_{\text{HJ}}$ methods with our ablations above baselines for the 7D Fixed-Wing plane system.

Training with $\mathcal{L}_{\text{CBF}}^2$ is ineffective, yielding nearly zero volume. The performance of NCBF severely declines due to the increased difficulty of sampling for counterexamples in higher dimensions. In contrast, DeepReach achieves higher volume at the cost of increased safety violations. Our method, PINN-CBF, achieves both low conservatism and safety. To illustrate this, we visualize the performance of PINN-CBF, NCBF, and DeepReach in filtering a nominal trajectory representing takeoff.

**Fixed-Wing Aircraft Example** In Fig. 3, the fixed-wing aircraft starting from the given position collides with a 3D obstacle, while in Fig. 4, obstacles are avoided with the PINN-CBF safety filter.

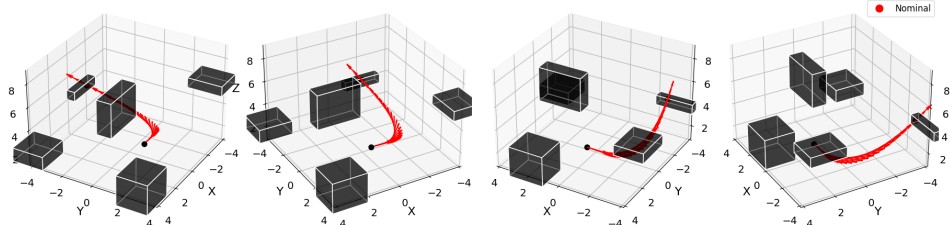

Figure 3: Nominal reference trajectory illustrating crash. The red arrow is the direction of velocity.

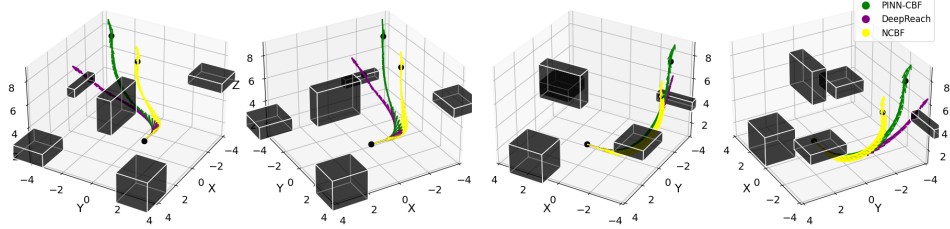

Figure 4: Agile takeoff avoiding obstacles under actuation limits: NCBF is conservative, DeepReach collides, PINN-CBF balances both. A video is linked here.

## 5 Conclusion and Limitations

We introduced a self-supervised framework for learning control barrier functions (CBFs) for limited-actuation systems with complex safety constraints. Our approach maximizes the control invariant set volume through physics-informed learning on a Hamilton-Jacobi (HJ) PDE characterizing the viability kernel. Additionally, we proposed a neural CBF parameterization that leverages the PDE structure.

A key limitation of learning-based CBFs is the lack of guarantees on the control invariant set, as well as the need for many samples, similar to physics-informed methods. Our intelligent sampling (IS) strategy mitigates this to some extent, but the approach remains restricted to specific domains and obstacle configurations. Future work will explore domain-adaptive CBFs that generalize across different obstacle settings, building on recent advances in physics-informed techniques and further investigating links between CBF learning, HJ reachability analysis, and Control Lyapunov Value Functions [44].

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

# A Appendix

## A.1 Reference Control

The reference control for the (DI) obeys the PID control law

$$u_{nom}(t) = K_p e(t) + K_i \int_0^t e(\tau)\, d\tau + K_d \frac{de(t)}{dt}$$

where $e(t)$ is the error between the current state and the target state at time $t$. For our experiments, we use the gains $K_p^x = -1.2$, $K_p^v = -2.0$, $K_d^x = 0.1$, $K_d^v = 0.1$ with $K_i^x = K_i^v = 0$.

For (F16), we hand-design a reference control policy resembling takeoff. To test the PINN-CBF filtering abilities, we obstruct the trajectory with rectangular obstacles and place walls to restrict the safe spatial coordinates to a box. In Figure 4, filtering demonstrates agile planning and obstacle avoidance. The following is an example control policy for takeoff while banking.

$$A_T(t) = \frac{a}{8} t + 1, \quad P(t) = \frac{p}{4} \sin(2t), \quad Q(t) = \frac{q}{5} \cos(t), \quad x(0) = [0, 1, 2, 0, 0, \pi, 1]$$

## A.2 3D Dubins Fixed-Wing Aircraft System Dynamics

Following [45], the dynamics of the Dubins fixed-wing aircraft (F16) system is given by:

$$
\begin{aligned}
&\dot{n} = V_T \cos\psi \cos\theta, && \dot{e} = V_T \sin\psi \cos\theta, \\
&\dot{d} = -V_T \sin\theta, && \dot{\phi} = P + \sin\phi \tan\theta\, Q + \cos\phi \tan\theta\, R, \\
&\dot{\theta} = \cos\phi\, Q - \sin\phi\, R, && \dot{\psi} = \frac{\sin\phi}{\cos\theta} Q + \frac{\cos\phi}{\cos\theta} R, \\
&\dot{V}_T = A_T, && R = \frac{g_D}{V_T} \sin\phi \cos\theta
\end{aligned}
\tag{12}
$$

where $[n\ e\ d\ \phi\ \theta\ \psi\ V_T]^\top$ are the states and $[A_T\ P\ Q]^T$ are the controls. In particular, $n, e, d$ are the position coordinates, $V_T$ is the tangential velocity of the aircraft, $\phi, \theta, \psi$ are the Euler coordinates orienting the plane, $A_T$ is the tangential acceleration control input, $P$ and $Q$ are rotational control inputs, and $g_D$ is gravitational acceleration.

