# OpenReview forum: "Learning Performance-oriented Control Barrier Functions Under Complex Safety Constraints and Limited Actuation"
_robot-learning.org/CoRL/2024/Conference — CoRL 2024_

### Official Review · Reviewer_EPGh · 2024-07-21
**Review of CoRL submission 404**

**Originality:** 2
**Technical Quality:** 3
**Clarity Of Presentation:** 3
**Potential Impact:** 2
**Recommendation:** 3
**Confidence:** 3

**Review:**

The paper is well-written and is easy to follow. Most of the notations and equations are self-contained and detailed. I like the CBF parameterization of Eq (10). But there are some weaknesses below.

- Some of the contributions seem to be incremental. The inner approximation is actually the log-sum-exp softmax in literature [14]. HJ PDE is also based on CBVF in previous work [32]. Moreover, in terms of PINN, the HJ loss in Eq. (12b) is the typical PDE residual error and it seems that there is no boundary condition loss to solve a well-posed problem, as it usually does in learning to solve PDE.

- I did not see the evidence of claimed "self-supervised training" because it seems that in Alg. 1, the pipeline still follows data collection and model training using ERM, which is typically supervised learning.

- In the experiments, Table 1 and Table 2 are confusing, and it is hard to tell which method performs the best. More importantly, I cannot find evidence that the proposed method significantly outperforms the baseline methods. It is expected to separate the ablation study with the comparison with baselines to better show the superiority of the proposed method.

- The fixed-wing aircraft example just gives a qualitative result and it is expected to have more quantitative results for comparison. Also, it is better to submit a video to show the results more clearly instead of Figure 7 and 8.

- Figure 1 needs more legends and notations to show the meaning of each component. There is a wrong reference on Line 231 with "??".

**Quality Of The Limitations Section:**

1

**Questions For Rebuttal:**

See weakness above.

**Robotics Focus:**

3

**Summary Of Paper:**

This paper proposes to learn a neural network control barrier function to achieve maximal performance by solving the HJ PDE using CBVF. To find the inner approximation of safe set, logical operators are adopted to construct state constraints. Experiments are conducted on two simulation examples.

**Summary Of Recommendation:**

Incremental contribution, unconvincing experiment, but more convincing after rebuttal

---

### Official Review · Reviewer_6drv · 2024-07-21

**Originality:** 2
**Technical Quality:** 3
**Clarity Of Presentation:** 4
**Potential Impact:** 3
**Recommendation:** 3
**Confidence:** 3

**Review:**

$\textbf{Quality and Clarity:}$ The paper is well-written, and the technical content is clear and concise. The reviewer appreciates the complete section on related works.

$\textbf{Originality:}$ The contributions of the paper include (1) the construction of a smooth minorizer of the constraint function, (2) a special parameterization of the neural CBF and an associated training scheme, and (3) the formulation of a self-supervised training loss that combines the HJ reachability and CBF objectives. (1) The authors note that the smooth composition of constraint functions is taken from [1], so the originality is unclear to the reviewer, apart from being applied in this specific framework. (2) As far as the reviewer understands, the proposed parameterization of the neural CBF and the proposed training scheme (e.g., initializing weights of last layer to zero, sampling training data only for nontrivial states), constitute a novel contribution. (3) The augmented training loss is only a slight modification of the well-established self-supervised HJ PDE loss used in, e.g., DeepReach [2], which lacks originality in the reviewer's opinion. Overall, the originality of the work is modest.

$\textbf{Significance:}$ The significance of the proposed smooth minorizer of the constraint function is unclear to the reviewer, since it is not compared to, e.g., the commonly used minimum signed distance function for HJ reachability problems in the experiments. The advantages of the proposed parameterization of the neural CBF is also unclear, since the results are not compared to existing neural baselines. The improvement achieved by using the proposed augmented training loss instead of the established HJ PDE loss is not clearly demonstrated by the experiment results.

$\textbf{Strengths:}$
 - The proposed framework is well-motivated by theoretical analysis of the structure of the CBVF.
 - The quality of the text and technical content is high.
 - The experiment results and analysis are well-presented.

$\textbf{Weaknesses:}$
 - The experiments section lacks comparisons to existing state-of-the-art baselines for synthesis of the CBVF.
 - The experiment results do not demonstrate a clear advantage achieved by the proposed modifications versus using the already established self-supervised HJ PDE loss.
 - The paper does not have a limitations section.

[1] Molnar, Tamas G., and Aaron D. Ames. "Composing control barrier functions for complex safety specifications." IEEE Control Systems Letters (2023).

[2] Bansal, Somil, and Claire J. Tomlin. "Deepreach: A deep learning approach to high-dimensional reachability." 2021 IEEE International Conference on Robotics and Automation (ICRA). IEEE, 2021.

**Quality Of The Limitations Section:**

1

**Questions For Rebuttal:**

- How does using the proposed smooth minorizer of the constraint function compare to using, e.g., the minimum signed distance function commonly used in HJ reachability problems?
 - The reviewer strongly suggests including a comparison of the proposed framework with an existing baseline to demonstrate the improvements achieved by the proposed parameterization of the neural CBF and proposed training scheme.
 - What key advantages does the CBVF formulation have compared against the HJ reachability value function formulation?
 - How are the algorithm hyperparameters $\lambda, \beta, \gamma$ selected?

**Robotics Focus:**

3

**Summary Of Paper:**

Synthesizing CBFs that maximize the resulting control invariant set for nonlinear high-dimensional systems remains an open challenge. The authors propose several useful modifications to existing learning-based reachability frameworks to stabilize and improve the training of a specially parameterized neural CBF. The efficacy of the proposed framework is demonstrated on a 2D double integrator system and a 7D fixed-wing aircraft system.

**Summary Of Recommendation:**

Although each proposed modification to the learning-based synthesis of the CBF is modest, the combined framework will likely serve as an illustrative baseline for future works. The paper is also greatly strengthened by high-quality writing, presentation, and context. For these reasons, I weakly recommend that the paper is accepted.

---

### Official Review · Reviewer_LoGn · 2024-07-21

**Originality:** 3
**Technical Quality:** 4
**Clarity Of Presentation:** 4
**Potential Impact:** 3
**Recommendation:** 3
**Confidence:** 4

**Review:**

Overall, the presentation of the paper is very clear but there are typos at multiple places. The motivation of the paper is also reasonable as there is a need of control barrier functions focused on improving performance of the system as CBFs usually tend to be conservative in nature and hamper system's performance. Until now, researchers have mostly focused on tuning class-K function to achieve this. The paper is theoretically sound and is the first to combine HJ reachability theory and control barrier functions theory when trying to learn neural network inspired control barrier functions. In table 1, the Neural Control barrier function (NCBF) method still outperforms the proposed method in terms of volume maximization which is a bit concerning as this was the main focus of the paper. Also, there are no hardware results for the proposed formulation. Since the CBF formulation is neural network inspired, 100% safety cannot be guaranteed and therefore the paper should address the safety rates for different formulations in Table 1 (see questions).

**Quality Of The Limitations Section:**

1

**Questions For Rebuttal:**

1. Since the paper utilizes a neural network based control barrier functions, 100% safety cannot be guaranteed. It would be great if the paper compares the safety rate of the different formulations listed in table 1. Does performance oriented CBF have lesser or more safety rate compared to other methods?

2. What effect the does the discount factor $\gamma$ have on the learned CBF. what value was chosen for implementation and was that kept the same throughout the training?

3. In section 3.2 what does each input $y_i$ correspond to. This is a bit confusing. They should clearly specify it and relate it to terms in section 3.1.

4. In Table 1, the volume of ${L_0}^{+}(h_{\theta})$ is higher for NCBF as compared to $\mathcal{L_{HJ}}$. Could you provide a reason for that?

5. The limitations of the proposed method are not mentioned. Please add that as well.

6. There are multiple typos in the paper which should be addressed.

**Robotics Focus:**

3

**Summary Of Paper:**

The paper introduces a novel approach towards learning control barrier functions focused on improving the performance of the system by maximizing the volume of the control invariant set. They utilize HJ-PDE inspired training loss to achieve this. Their formulation handles multiple safety constraints by utilizing composition of safety constraints followed by smoothening to get an inner approximation of the safe set.

**Summary Of Recommendation:**

Please read the review above.

---

### Author Rebuttal · Authors · 2024-08-13

| Method | Mean $\lvert \mathcal{L}_{\text{HJ}} \rvert$ over $\mathcal{D}$ | Mean $\mathcal{L}_{\text{CBF}}$ over $L_0^+(c)$ | $\mathrm{Vol}(L_0^+(h_\theta)) / \mathrm{Vol}(L_0^+(c))$ |
|--------|:-----------------------------------:|:----------------------------------:|:--------------------------------------------:|
| $\mathcal{L}_{\text{HJ}}$ | $0.045$ | $0.026$ | $0.520$ |
| $\mathcal{L}_{\text{CBF}}$ [Dawson et al., 2022] | $1.300$ | $9.093 \times 10^{-5}$ | $0.036$ |
| $\mathcal{L}_{\text{CBF } + \text{HJ}}$ | $0.050$ | $0.022$ | $0.480$ |
| NCBF [Liu et al., 2023] | $0.262$ | $0.138$ | $0.955$ |
| DeepReach [Bansal et al., 2021] | $0.302$ | $0.194$ | $0.943$ |

Table 1:Comparison of $\mathcal{L}_{\text{HJ}}$ methods with baselines for the 2D Double-Integrator system.

| Model | Mean $\lvert \mathcal{L}_{\text{HJ}} \rvert$ over $\mathcal{D}$ | Mean $\mathcal{L}_{\text{CBF}}$ over $L_0^+(c)$ | $\mathrm{Vol}(L_0^+(h_\theta)) / \mathrm{Vol}(L_0^+(c))$|
|-------|:-------------------------------:|:----------------------------------:|:--------------------------------------------:|
| $\mathcal{L}_{\text{HJ}} $ | $0.373$ | $0.073$ | $0.600$ |
| $\mathcal{L}_{\text{CBF}}$ [Dawson et al., 2022] | $3.267$ | $8.654 \times 10^{-5}$ | $0.081$ |
| $\mathcal{L}_{\text{CBF } + \text{HJ}}$| $0.398$ | $0.065$ | $0.567$ |
| NCBF [Liu et al., 2023] | $0.782$ | $0.478$ | $0.809$ |
| DeepReach [Bansal et al., 2021] | $0.732$ | $0.477$ | $0.866$ |
**Table 2:** Comparison of $\mathcal{L}_{\text{HJ}}$ methods with baselines for the 7D Fixed-Wing plane system.

### Limitations

- Being a learning-based approach, our method lacks guarantees on the control invariant set; however, our parameterization ensures that the learned invariant set is free of obstacles.
- There can be places within the learned invariant set where the volume of feasible controls is very small. This is referred to as input saturation and is illustrated in Figures 3 & 4 through the calculation of $\mathcal{K}_{\text{CBF}}$ over the domain.
- Finally, our current approach applies to a fixed domain and obstacle configuration. If there is any significant change, the model will have to be retrained.

---

### Decision · Program_Chairs · 2024-09-04

**Decision:**

Accept

**Comment:**

The paper introduces an approach to learning control barrier functions focused on improving the performance of the system by maximizing the volume of the control invariant set. They utilize HJ-PDE inspired training loss to achieve this. The proposed formulation handles multiple safety constraints by utilizing a composition of safety constraints followed by smoothening to get an inner approximation of the safe set.

Strengths
- The paper is well-written and easy to follow. The technical content is clearly presented.
- The proposed framework is well-motivated by theoretical analysis of the structure of the CBVF.

Limitations

- Several reviewers have raised concerns regarding the novelty and contribution of the proposed approach over existing methods.

- The experiment results do not demonstrate a clear advantage achieved by the proposed modifications over existing approaches. Further quantitative comparison is required to establish the utility of the proposed approach.

- The experiment results presented as it is somewhat confusing. It will help to separate out the baseline comparisons and ablation studies, clearly marking the winning approach.

- The authors should discuss the limitations of the proposed approach, such as the effect of using a learning-based approach for safety.

**Comments post rebuttal**

The authors have successfully addressed reviewers' concerns. The new experiments conducted by the authors were particularly useful in demonstrating the utility of the proposed method. The authors should incorporate the additional experiment results into the revised paper to make it more solid and convincing. In addition, the authors should include a metric for comparison that is more interpretable than a loss term (e.g., some sort of % safety rate?) in the final version.